# Learning Multi-Level Features with Matryoshka Sparse Autoencoders

**Bart Bussmann** [*]  **Noa Nabeshima** [*]  **Adam Karvonen**  **Neel Nanda**

## Abstract

Sparse autoencoders (SAEs) have emerged as a powerful tool for interpreting neural networks by extracting the concepts represented in their activations. However, choosing the size of the SAE dictionary (i.e. number of learned concepts) creates a tension: as dictionary size increases to capture more relevant concepts, sparsity incentivizes features to be split or absorbed into more specific features, leaving high-level features missing or warped. We introduce Matryoshka SAEs, a novel variant that addresses these issues by simultaneously training multiple nested dictionaries of increasing size, forcing the smaller dictionaries to independently reconstruct the inputs without using the larger dictionaries. This organizes features hierarchically - the smaller dictionaries learn general concepts, while the larger dictionaries learn more specific concepts, without incentive to absorb the high-level features. We train Matryoshka SAEs on Gemma-2-2B and TinyStories and find superior performance on sparse probing and targeted concept erasure tasks, more disentangled concept representations, and reduced feature absorption. While there is a minor trade-off with reconstruction performance, we believe Matryoshka SAEs are a superior alternative for practical tasks, as they enable training arbitrarily large SAEs while retaining interpretable features at different levels of abstraction.

## 1. Introduction

Sparse autoencoders (SAEs) have emerged as a powerful tool for interpreting the internal representations of neural networks (Bricken et al., 2023; Cunningham et al., 2023). By training autoencoders to map dense activations of neural networks into a *sparse* latent space, researchers aim to

extract meaningful, human-interpretable features that provide insight into a model's decision-making process. In practice, however, sparsity is only an imperfect proxy for interpretability; optimizing to represent inputs using as few active latents as possible can lead to pathological solutions.

A core issue lies in the mismatch between the flat sparsity objective and the inherent hierarchical structure of real-world features. For example, a broad, general concept (e.g. "punctuation marks") might be a parent feature to more specific ones ("punctuation mark and period", "punctuation mark and question mark," and "punctuation mark and comma"). When an SAE has the capacity to learn many latents, the sparsity penalty can push it to replace each general concept with a set of narrowly specialized features without retaining the original high-level category—a phenomenon called *feature splitting* (Bricken et al., 2023). Even more problematic is *feature absorption* (Chanin et al., 2024), where a parent feature only partially splits and a number of specific instances of a general feature are absorbed by more specialized latents, leaving "holes" in the representation of the more general feature (e.g. a latent that activates on all tokens starting with an E, except if the word is Elephant). Finally, *feature composition* occurs when the SAE merges distinct concepts (like "red" and "triangle") into single composite features ("red triangle") to minimize the number of active latents, instead of learning the underlying features (Anders et al., 2024; Wattenberg & Viégas, 2024; Leask et al., 2025).

These problems become more severe as SAEs scale to larger dictionary sizes (i.e., more learned concepts), since the model exploits the additional capacity to further minimize active latents through mechanisms like feature splitting (Karvonen et al., 2024a). This presents a stark contrast to typical deep learning, where scaling consistently improves both training loss and model capabilities (Kaplan et al., 2020; Hoffmann et al., 2022). While larger SAEs do achieve lower training loss (Gao et al., 2024; Templeton, 2024), their performance on downstream tasks often deteriorates (Karvonen et al., 2024a). This fundamental tension creates significant practical challenges for deploying SAEs in interpretability research - they become less reliable for probing model behavior, steering models, analyzing feature circuits, or understanding how models process information. The current recommended approach is to train a sweep of different sizes and evaluate each of them, but this is costly (Lieberum et al.,

[*]Equal contribution . Author order determined by coin flip. Correspondence to: Bart Bussmann <bartbussmann@gmail.com>.

*Proceedings of the 42$^{nd}$ International Conference on Machine Learning*, Vancouver, Canada. PMLR 267, 2025. Copyright 2025 by the author(s).

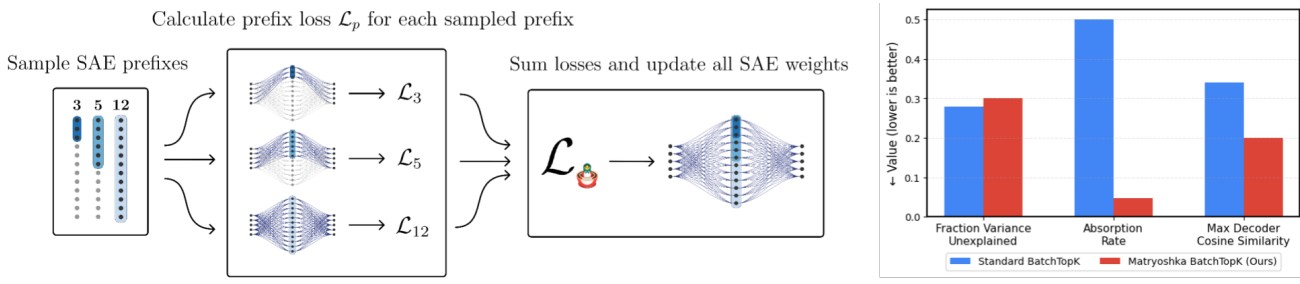

*Figure 1.* **Architecture and Performance of Matryoshka SAEs.** (Left) The model learns multiple nested reconstructions simultaneously, with each reconstruction using only a subset of the total latents. This creates pressure for early latents to capture general features while later latents can specialize in more specific concepts. (Right) Comparative metrics between SAEs with average sparsity of 40 active latents per token (L0=40) showing that while Matryoshka SAEs sacrifice a small amount of reconstruction fidelity (higher variance unexplained), they achieve significantly lower feature absorption rates and less feature composition (lower decoder cosine similarity).

2024). Ideally, we would like to have a single SAE that can capture both high-level and fine-grained features.

Therefore, we introduce **Matryoshka SAEs**, a novel hierarchical approach to SAE training, named after the Russian nesting dolls and inspired by Kusupati et al. (2024). Matryoshka SAEs learn a single feature space that retains both abstract and specific features by training multiple nested SAEs of increasing size simultaneously. Each nested sub-SAE is optimized to reconstruct the input using only a subset of the total latents (including the sub-SAEs nested within itself). This nested structure prevents later, more specialized latents from absorbing the roles of earlier, more general ones, effectively regularizing the SAE to learn features at multiple levels of abstraction. Despite the need for reconstructions at multiple scales, Matryoshka SAEs can be implemented efficiently with a modest increase in training time compared to traditional SAEs, depending on the number of nested sub-SAEs.

Through extensive experiments on both synthetic and real-world datasets, we demonstrate that Matryoshka SAEs mitigate feature absorption across a wide range of model sizes and sparsity levels. We find that Matryoshka SAEs have more disentangled latent representations (as measured by maximum cosine similarity of decoder vectors) and also improve performance on probing and targeted concept erasure tasks compared to standard SAE baselines. Furthermore, we find that with increasing dictionary size, Matryoshka SAEs often improve or maintain their performance on downstream tasks, where other architectures deteriorate in performance. While Matryoshka SAEs exhibit a small trade-off in reconstruction error due to the additional constraints, we argue that the improved quality of the learned latents, which also get better with scaling, make them an attractive alternative for practical applications.

We consider our results a positive sign for the use of SAEs

in interpretability research. We find that concerning failure modes that were thought to be fundamental barriers to using SAEs, can be mitigated through simple changes in the training procedure. This indicates that other problems with SAEs might be easier to solve than thought, and illustrates the promise of more research in this area.

## 2. Background

### 2.1. Interpretability with Sparse Autoencoders

Early mechanistic interpretability work sought to understand neural networks by examining individual architectural components, but faced challenges due to polysemanticity - where components seem to exhibit multiple unrelated behaviors (Olah et al., 2020; Elhage et al., 2022). In recent years, SAEs emerged as a promising solution for extracting monosemantic and interpretable features from model activations (Bricken et al., 2023; Cunningham et al., 2023; Templeton, 2024; Gao et al., 2024; Rajamanoharan et al., 2024a;b), which researchers can analyze through manual inspection (Bricken et al., 2023; Lin, 2023) or automated interpretability techniques (Gao et al., 2024; Juang et al., 2024). Their effectiveness has been demonstrated across various applications, such as analyzing attention mechanisms in GPT-2 (Kissane et al., 2024), circuit analysis (Marks et al., 2024; Makelov et al., 2024), and model-diffing (Lindsey et al., 2024).

Technically, an SAE consists of an encoder and a decoder:

$$\mathbf{f}(\mathbf{x}) = \sigma(\mathbf{W}^{\text{enc}}\mathbf{x} + \mathbf{b}^{\text{enc}}) \tag{1}$$

$$\hat{\mathbf{x}} = \mathbf{W}^{\text{dec}}\mathbf{f}(\mathbf{x}) + \mathbf{b}^{\text{dec}} \tag{2}$$

The encoder is trained to map high-dimensional activations or hidden states from inside a neural network into a sparse,

overcomplete representation $\mathbf{f}(\mathbf{x})$, called the SAE *latents* or *features*. The activation function $\sigma$ enforces non-negativity and sparsity in $\mathbf{f}(\mathbf{x})$, with a latent $i$ typically considered active when $f_i(\mathbf{x}) > 0$.

The decoder maps the sparse features back to the original input space through a linear transformation, producing an approximate reconstruction $\hat{\mathbf{x}}$. This reconstruction aims to minimize the mean squared error with respect to the original input $\mathbf{x}$ while maintaining the sparsity constraints on the intermediate representation.

Recent architectural improvements include JumpReLU SAEs (Rajamanoharan et al., 2024b), which learn individual activation thresholds, and TopK SAEs (Gao et al., 2024), which enforce exactly $K$ active features per sample. For our Gemma-2-2B experiments, we use the BatchTopK (Bussmann et al., 2024) activation function, which we further describe in Section 3.2.

## 2.2. Challenges in Scaling SAEs

At the heart of SAE training lies a fundamental tension between reconstruction quality and sparsity. The sparsity objective, typically implemented via L1 regularization or explicit L0 constraints such as TopK, encourages SAEs to represent inputs using as few active latents as possible. While some degree of sparsity is necessary to extract interpretable features from superposition, optimizing for the reconstruction + sparsity loss often leads to pathological solutions. This manifests in several interrelated phenomena:

**Feature splitting** (Bricken et al., 2023) causes unified concepts to fragment into many specialized latents – for example, a feature responding to punctuation marks may split into separate features for question marks, periods, and commas. Even though each individual, specialized latent might be interpretable, the high-level concept of "punctuation mark" - which might be functionally used by the LLM - is now missing from the dictionary. Instead, the "punctuation mark" decoder direction is now implicitly represented in the decoder vector of each specific latent, as for example "comma" also implies "punctuation mark". Leask et al. (2025) found that when doubling the dictionary size of an SAE, only one-third of the latents in the larger SAE represent novel concepts, while the other latents represent concepts similar to those of the smaller SAE in a sparser way.

**Feature absorption** (Chanin et al., 2024) occurs when a latent representing a general feature develops systematic blind spots in cases handled by more specialized latents. For example, consider a latent that activates on female names like "Mary", "Jane", "Sarah", and "Lily". If a specialized latent splits for the name "Lily", the latent for female names might instead become an "all female names except Lily" latent. Unlike feature splitting where general concepts completely

fragment, absorption creates specific "holes" in otherwise intact general features, making the general feature less reliable for downstream tasks and harder to interpret.

**Feature composition** represents another sparsity-driven distortion that occurs when features naturally co-occur. When presented with a direct product space of independent features (e.g. colors and shapes), the sparsity objective incentivizes learning single latents that capture specific combinations (like "red triangle") rather than representing the underlying independent features ("red" and "triangle") separately (Anders et al., 2024; Wattenberg & Viégas, 2024). Even when these features are conceptually and functionally independent in the model, combining them allows the SAE to achieve the same reconstruction with fewer active latents. Leask et al. (2025) have shown using "meta-SAEs" that the latents of sparse autoencoders are often composed of more fundamental "meta-latents".

These issues become more severe as dictionary size increases. While larger dictionaries allow for representing more relevant features, they also enable more opportunities for sparsity-driven distortions. The reconstruction objective alone does not prevent these distortions since they achieve similar reconstruction with fewer active latents. Their persistence suggests they arise from fundamental limitations in standard SAE training objectives rather than optimization challenges. This creates a pressing need for new training approaches that can preserve features at multiple levels of abstraction while maintaining the benefits of sparsity. Ideally, we want SAEs that can capture both high-level concepts and their refinements without letting specialized features absorb or fragment their general counterparts.

## 2.3. Matryoshka Representation Learning

Matryoshka Representation Learning (MRL) (Kusupati et al., 2024) is an approach to representation learning that encodes information at varying levels of granularity within a single embedding vector. This allows the representation to be adaptable to the computational constraints of diverse downstream tasks. MRL is designed to modify existing representation learning pipelines with minimal overhead during training and no additional costs during inference and deployment. The key distinction of MRL lies in its loss function, which is modified to incorporate the performance when using only part of the embedding vector. The name "Matryoshka" is inspired by the nested Russian dolls, reflecting the nested nature of the representations.

## 3. Matryoshka Sparse Autoencoders

Matryoshka SAEs extend traditional sparse autoencoders by simultaneously training multiple nested autoencoders of increasing size, as illustrated in Figure 1. Given a maximum

dictionary size $m$, we define a sequence of nested dictionary sizes $\mathcal{M} = m_1, m_2, ..., m_n$ where $m_1 < m_2 < ... < m_n = m$. Each size $m_i$ corresponds to a sub-SAE that must reconstruct the input using only the first $m_i$ latents.

We experiment with two types of Matryoshka SAE: *random prefix*, where $\mathcal{M}$ is randomly sampled from a distribution per batch, and *fixed prefix* where $\mathcal{M}$ is set as a hyperparameter before training. Formally, for an input $\mathbf{x} \in \mathbb{R}^n$, the encoder and decoder are defined as:

$$\mathbf{f}(\mathbf{x}) = \sigma(\mathbf{W}^{\text{enc}}\mathbf{x} + \mathbf{b}^{\text{enc}}) \tag{3}$$

$$\hat{\mathbf{x}}_i(\mathbf{f}) = \mathbf{W}^{\text{dec}}_{0:m_i,:}\mathbf{f}_{0:m_i} + \mathbf{b}^{\text{dec}} \quad \text{for } m_i \in \mathcal{M} \tag{4}$$

where $\mathbf{W}^{\text{enc}} \in \mathbb{R}^{m \times n}$ is the encoder matrix, $\mathbf{W}^{\text{dec}} \in \mathbb{R}^{n \times m}$ is the decoder matrix, $\mathbf{b}^{\text{enc}} \in \mathbb{R}^m$ is the encoder bias, $\mathbf{b}^{\text{dec}} \in \mathbb{R}^n$ is the decoder bias, and the subscript notation $0 : m_i$ indicates taking the first $m_i$ rows or elements. Each nested decoder $\mathbf{W}^{\text{dec}}_{0:m_i,:}$ must learn to reconstruct the input using only a subset of the latents, creating a hierarchy of representations at different scales.

### 3.1. Training Objective

The key innovation in Matryoshka SAEs is the training objective that enforces good reconstruction at multiple scales simultaneously:

$$\mathcal{L}(\mathbf{x}) = \sum_{m \in \mathcal{M}} \|\mathbf{x} - \underbrace{(\mathbf{f}(\mathbf{x})_{0:m}\mathbf{W}^{\text{dec}}_{0:m} + \mathbf{b}^{\text{dec}})}_{\text{reconstruction using first } m \text{ latents}} \|_2^2 + \alpha\mathcal{L}_{\text{aux}} \tag{5}$$

where $\mathcal{L}_{\text{aux}}$ is the standard auxilary loss as used in Gao et al. (2024). The first $m_1$ latents must learn to reconstruct the input as well as possible on their own, the first $m_2$ latents must do the same, and so on. Early latents must be able to reconstruct the input when $m_i$ is smaller, creating pressure for them to capture general, widely applicable features. Later latents only participate in reconstructing larger dictionaries, allowing them to specialize in more specific features.

### 3.2. BatchTopK Activation Function

For our Gemma-2-2B language model experiments, we use the BatchTopK activation function (Bussmann et al., 2024) to determine which latents are active. The BatchTopK activation function improves upon standard element-wise approaches by considering sparsity across batches rather than individual examples. For a batch $\mathbf{X} = [\mathbf{x}_1, ..., \mathbf{x}_B]$, BatchTopK retains the $B \times K$ largest activations across the entire batch while setting others to zero:

$$\text{BatchTopK}(\mathbf{X}) = \mathbf{X} \odot \mathbf{1}[\mathbf{X} \geq \tau(\mathbf{X})] \tag{6}$$

where $\tau(\mathbf{X})$ is the $(B \times K)$th largest value in $\mathbf{X}$, $\odot$ denotes element-wise multiplication, and $\mathbf{1}[\cdot]$ is the indicator function. This allows the number of active latents per example to vary naturally while maintaining an average sparsity of $K$ across the batch. During inference, BatchTopK is replaced with a global threshold to ensure consistent behavior independent of the batch:

$$\text{BatchTopK}(\mathbf{x}) = \mathbf{x} \odot \mathbf{1}[\mathbf{x} \geq \theta] \tag{7}$$

where $\theta$ is calibrated on the training data to maintain the desired average sparsity.

## 4. Experiments

To understand how Matryoshka SAEs prevent feature absorption and fragmentation, we conduct experiments across three settings of increasing complexity: a synthetic toy model designed to exhibit feature absorption, a qualitative investigation of the features learned on a small 4-layer TinyStories model, and finally benchmarks on the larger Gemma-2-2B architecture.

### 4.1. Toy Model Demonstration of Feature Absorption

We first demonstrate the ability of Matryoshka SAEs to avoid feature absorption in a controlled, synthetic setting, with a toy model following the model introduced in Chanin et al. (2024). Concretely, we construct a tree-structured set of $L$ binary features, each mapped to a unique direction in $\mathbb{R}^d$ (we use $d = 20$). The root feature is always present (though it is excluded from $L$). Every other feature is sampled conditional on its parent being present, with an associated edge probability $p_{(\text{parent} \rightarrow \text{child})} \in (0, 1)$. If a feature is active, we add its direction vector to produce the final $d$-dimensional input. This induces a hierarchical dependency analogous to "comma" $\implies$ "punctuation mark" in real text: child features *always* appear with their parent features.

We train two types of autoencoders on data generated from this model: a standard sparse autoencoder (*Vanilla SAE*) with $L$ latents, matching the true number of features and a *Matryoshka SAE* with the same total latents $L$, but with additional nested reconstruction objectives on sub-prefixes of the latents. For the Matryoshka SAE, we sample the prefix lengths from a truncated Pareto distribution at each training step, and use a ReLU activation function (see Appendix C for more training details). Both models have the same encoder and decoder architecture.

Figure 3 shows the ground-truth features against the closest-

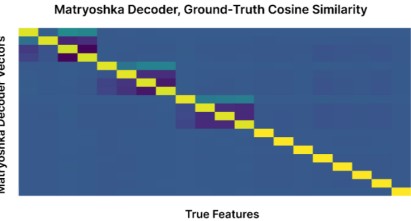
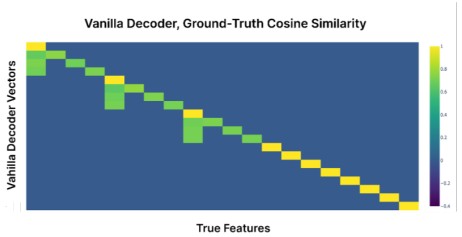

*Figure 2.* **Toy model decoder vector similarity.** Graphical representation of the toy model (left). The heatmaps show the cosine similarity between learned latent vectors and ground-truth feature vectors for the Matryoshka SAE (middle) and Vanilla SAE (right). The Matryoshka SAE shows a clear diagonal structure, which demonstrates disentanglement of the hierarchical features and learning the ground truth. The Vanilla SAE, however, exhibits high similarity between parent and child latents, indicating feature absorption.

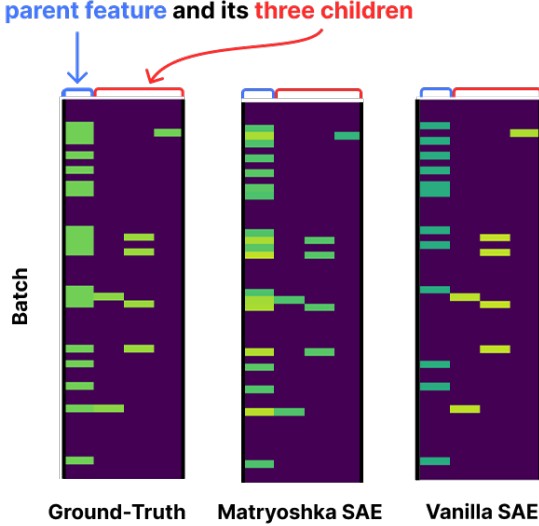

*Figure 3.* **Toy model activations.** Ground-truth feature activations alongside Matryoshka SAE activations and Vanilla activations for one of the parents and its children on the toy model. Notice that in the Vanilla SAE the parent latent (column bracketed in blue) does not fire when its children (bracketed in red) fire. For the activations of all features in the toy model, see Figure 11.

matching Matryoshka and Vanilla activations on a single batch for a single parent and its children features (for a plot of the activations of all features see Figure 11). The Vanilla SAE exhibits feature absorption - when one of the child features is active, the corresponding parent feature does not activate. In contrast, the Matryoshka SAE recovers the true underlying feature structure, with parent features remaining active regardless of which child features are active.

To further see whether the autoencoders have found the ground truth features, we measure the cosine similarity between the learned decoder vectors and the ground-truth feature vectors (Figure 2). For the Vanilla SAE, we observe high similarity between parent and child latents, indicating

that the model is learning redundant representations and failing to disentangle the hierarchical features. The Matryoshka SAE, on the other hand, shows a clear diagonal structure in its similarity matrix, demonstrating that it is able to recover the true feature vectors with minimal overlap, with only small nonzero off-diagonal terms.

### 4.2. TinyStories Investigation

We now examine feature absorption in a 4-layer Transformer[1] (hidden size 768) trained on TinyStories (Eldan & Li, 2023), a dataset of simple English children's stories. To systematically study how features evolve as SAE dictionary size increases, we train a family of small "reference" SAEs with varying dictionary sizes (30, 100, 300, 1k, 3k, and 10k latents) on three model locations: attention block outputs, MLP outputs, and the residual stream.

These reference SAEs reveal a consistent pattern: smaller SAEs (like S/1 with 100 latents) often capture broad, general concepts, while larger ones tend to fragment these concepts into specialized cases. To visualize these relationships, we developed an interactive tool (sparselatents.com/tree_view) that traces how latents evolve across SAE sizes. Technical details about the visualization methodology can be found in Appendix E.

**"Female tokens" Absorption Example.** In Figure 4, we highlight latent 65 in our 300-latent reference SAE (S/2/65), which consistently fires on female-coded tokens: "she," "her," "girl," and female names like "Lily" and "Sue." When we examine the corresponding latents in the 1000-latent SAE (S/3), we find that the general female-words latent (S/3/66) has developed systematic "holes"—it stops firing on "Lily" because a specialized "Lily-specific" latent (S/3/359) has absorbed that case. The original concept becomes fragmented, obscuring that the model treats "Lily" as a female name.

---

[1] https://github.com/noanabeshima/tinymodel

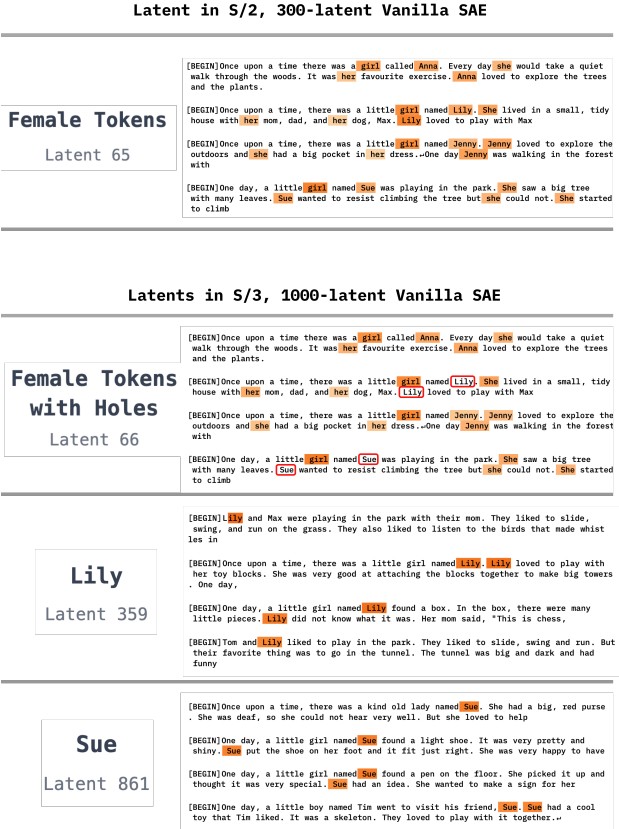

*Figure 4.* **Feature absorption in a TinyStories model.** Example activations showing how a general "female words" latent (S/2/65) develops holes in a larger SAE (S/3/66) as specialized latents for "Lily" (S/3/359) and "Sue" (S/3/861) absorb specific cases. Feature absorption locations are circled in red.

In contrast, when we train a 25k-latent Matryoshka SAE with the same average sparsity, it maintains the broad coverage of female-coded tokens while still developing specialized child latents (e.g. a Sue latent, Lily latent, and a female names latent without holes, see here). This preserves the hierarchical relationship between general and specific features, making the model's representations easier to interpret. In Appendix D, we show a few examples of features learned by the Matryoshka SAE.

### 4.3. Larger-Scale Validation with Gemma-2-2B

In our Gemma-2-2B language model experiment, we use fixed prefix Matryoshka over random as we found that they attained slightly better eval metrics (see Appendix H for details). We use a dictionary size of $D = 65536$ with five nested sub-SAEs where $\mathcal{M} = \{2048, 6144, 14336, 30720, 65536\}$. We train five Matryoshka SAEs with an average sparsity of respectively 20, 40, 80, 160, and 320 active latents per token.

The SAEs were trained on the residual stream activations from layer 12 of Gemma 2-2B using 500M tokens sampled from The Pile (Gao et al., 2020). We use the Adam optimizer with a learning rate of $3 \times 10^{-4}$ and batch size of 2048. No additional regularization terms are used beyond the implicit regularization from the multiple reconstruction objectives.

We evaluate Matryoshka SAEs against six alternative SAE architectures (see Appendix A) to assess both basic performance and effectiveness in addressing feature absorption and composition challenges. We use the evaluations provided by SAE Bench (Karvonen et al., 2024a) to compare Matryoshka SAEs to the baseline SAEs across a wide variety of tasks. In these evaluations, we use all latents of the Matryoshka SAE. In Appendix F, we study the performance of the different nested sub-SAEs.

**Reconstruction and Downstream Performance.** We first examine basic performance metrics of the SAEs: reconstruction quality (measured by variance of the input explained by the reconstruction) and downstream performance (measured by cross-entropy loss when feeding reconstructed activations back into the language model). Figure 5 shows these metrics across different sparsity levels.

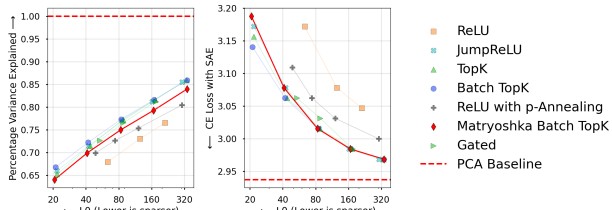

*Figure 5.* **Reconstruction performance.** The variance explained of Matryoshka SAEs is slightly worse than some competing architectures. However, looking at the downstream LLM CE loss, Matroshka SAEs perform comparable, especially at larger L0s.

Matryoshka SAEs consistently show slightly worse reconstruction compared to standard BatchTopK SAEs. For example, at a L0 of 40 the reconstruction of Matryoshka SAEs explains 70% of the variance in model activations, whereas BatchTopK SAEs explain 72% of the variance. This performance gap is expected, as the nested reconstruction constraints prevent Matryoshka SAEs from splitting and absorbing broad latents to optimize sparsity. Therefore, it would be surprising if there was not mild degradation here. The PCA baseline, included for reference, is not subjected to any sparsity constraints and can therefore utilize its full capacity for reconstruction, unlike the SAEs.

However, downstream model loss tells a different story. Despite worse reconstruction, at larger L0s, Matryoshka SAEs achieve comparable cross-entropy loss when their recon-

structed activations are fed back into the language model. This suggests that Matryoshka SAEs may be learning more meaningful features that better capture the language model's internal representations, even though they reconstruct the raw activations less accurately.

**Feature Absorption and Splitting.** We evaluate the extent to which Matryoshka SAEs reduce two key pathologies that emerge when scaling sparse autoencoders: feature absorption and feature splitting. Following the methodology of Chanin et al. (2024), we use first-letter classification tasks as a probe to measure these phenomena.

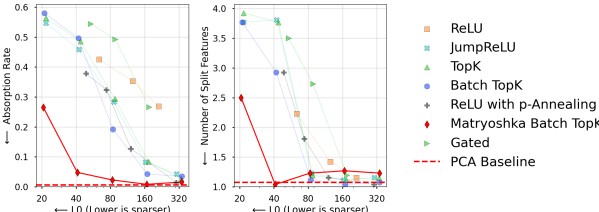

*Figure 6.* **Feature absorption and splitting**. Matryoshka SAEs significantly reduce feature absorption and splitting compared to baselines.

The absorption rate, as reported in Figure 6 and calculated following SAEBench (Karvonen et al., 2024a), measures the fraction of tokens where a latent corresponding to a first-letter feature fails to activate despite the token starting with that letter, because the feature is absorbed by another latent. To quantify feature absorption, we first train logistic regression probes on the raw activations to establish a ground truth for which directions encode first-letter information. The absorption rate is then determined by finding false-negative tokens where main SAE latents (identified via k-sparse probing for the letter feature) fail to fully activate (their projection onto the ground truth direction is less than the model's projection) while other "absorbing" latents compensate on these tokens. For further details on this metric, we refer to Chanin et al. (2024) and the SAEBench implementation.

Secondly, we also analyze feature splitting. The splitting metric in Figure 6, also from SAEBench, counts how many latents are needed to represent a single first-letter feature. We measure this by training k-sparse probes on SAE latents and detecting when increasing $k$ (the number of latents used by the probe) by one causes a jump in F1 score by more than a threshold $\tau = 0.03$. Such a jump indicates that additional latents contain significant information about the feature, suggesting the feature has been split across those multiple latents.

As shown in Figure 6, Matryoshka SAEs reduces both fea-

ture absorption and splitting compared to the baseline architectures. For example, at an average sparsity (L0) of 40 active latents per token, Matryoshka SAEs exhibit an absorption rate of just 0.05, in contrast to 0.49 for BatchTopK SAEs. This indicates that Matryoshka SAEs are far more successful at preserving general first-letter features in the presence of specialized latents.

Similarly, Matryoshka SAEs require fewer latents to capture first-letter features, demonstrating reduced feature splitting. At an L0 of 40, Matryoshka SAEs need only one latent per first letter on average, while BatchTopK SAEs split this information across three latents.

These results demonstrate that the hierarchical structure imposed by Matryoshka SAEs is effective at combating the feature absorption and splitting that typically emerge when increasing SAE dictionary size. Matryoshka SAEs are able to maintain coherent high-level features even in the presence of specialized low-level features.

**Sparse Probing and Targeted Concept Removal.** We evaluate the quality of the features learned by the SAEs through three complementary analyses that measure different aspects of feature quality and disentanglement.

First, we perform sparse probing to assess whether the learned latents encode semantically meaningful concepts (Gao et al., 2024). We use a suite of 35 binary classification tasks spanning five domains: professions, sentiment, language, code, and news. For each task, we encode the inputs through the SAE, apply mean pooling over non-padding tokens, select the most relevant latent, and train a logistic regression probe. Strong performance on these tasks indicates that the SAE has learned relevant latents and are not missing, entangled, or split across multiple latents.

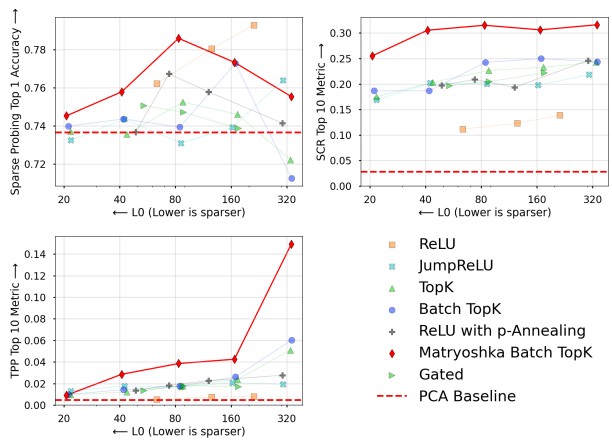

*Figure 7.* **Probing and concept isolation**. Matryoshka SAEs score considerably better at SCR and TPP than baseline models, and better at sparse probing at lower sparsities.

Second, we test the SAE's ability to separate dataset biases, such as the spurious correlation between a profession like "nurse" always being associated with "female", using spurious correlation removal (SCR) (Karvonen et al., 2024b). We train classifiers on intentionally biased data, then ablate the latents that encode the spurious signal. A higher resulting accuracy suggests the SAE was more effective at isolating the spurious correlation (e.g. gender), enabling the classifier to focus on the intended task (e.g. profession classification).

Third, we evaluate how well the SAE isolates individual concepts using Targeted Probe Perturbation (TPP) (Karvonen et al., 2024b). We ablate latents specific to one class and measure the change in accuracy of a classifier trained for that class as well as classifiers for other classes. If the ablated latents are properly disentangled, it should have an isolated causal effect - reducing accuracy on the relevant class probe while leaving other class probes unaffected.

As shown in Figure 7, Matryoshka SAEs significantly outperform all benchmark architectures on the TPP and SCR metrics, demonstrating they learn more disentangled and isolated concept representations. On the sparse probing tasks, Matryoshka SAEs achieve the best performance at lower sparsity levels and consistently outperform standard BatchTopK SAEs across all sparsity levels. These results strongly suggest that the hierarchical structure imposed by Matryoshka SAEs leads to higher quality and more relevant learned features.

**Feature Composition Analysis.** To quantify feature composition and shared information between latents, we look at the average maximum cosine similarity between two latents in an SAE. If the SAE has many latents with high cosine similarity, this means that multiple latents are representing similar information, indicating composition of information.

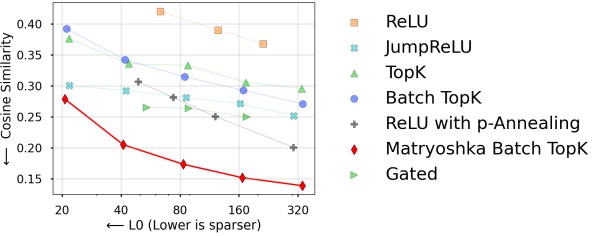

*Figure 8.* **Average maximum cosine similarity**. The maximium cosine similarity is considerably lower for Matryoshka SAEs, indicating less feature composition.

In Figure 8, we see that the average maximum cosine similarity between decoder vectors is substantially lower for Matryoshka SAEs, indicating feature composition is less of a problem for Matryoshka SAEs. In Appendix F.2, we investigate this phenomenon further using meta-SAEs and find that the same holds.

**Automated Interpretability.** We evaluate feature interpretability using gpt4o-mini as an LLM judge. Following Bills et al. (2023) and Paulo et al. (2024), we generate feature explanations for 1000 latents from a range of activating dataset examples and score the explanation by asking an LLM judge to predict on which inputs a latent will be active. Figure 9 shows that the interpretability of latents of Matryoshka SAEs are comparable to standard BatchTopK SAEs and more interpretable than many alternative baseline architectures.

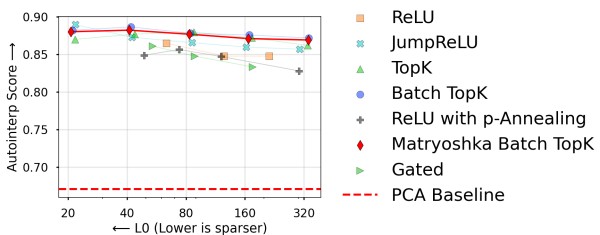

*Figure 9.* **Automatic interpretability score**. Matryoshka SAE latents are among the most interpretable.

**Scaling SAEs** To evaluate how Matryoshka SAEs performance changes when increasing dictionary size, we train Matryoshka SAEs with dictionary sizes of 4k, 16k, and 65k. Figure 10 shows that on almost all metrics Matryoshka SAEs improve or are stable with scale, whereas alternative architectures often degrade. These results indicate that as one increases the dictionary size of the SAE to capture more of features of the LLM, Matryoshka SAEs may be the superior choice for many downstream tasks.

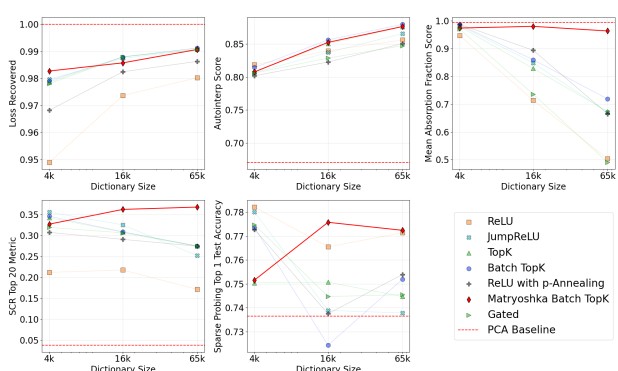

*Figure 10.* **Scaling evaluations**. Performance across three different dictionary sizes. Average performance across L0s 40, 80, 160 are reported.

## 5. Limitations

While Matryoshka SAEs demonstrate improved performance of downstream tasks, this comes at the cost of slightly reduced reconstruction performance compared to standard SAEs. Although downstream loss remains competitive, applications requiring precise activation reconstruction may find this trade-off problematic.

Furthermore, the multiple nested reconstruction objectives increases training time compared to standard SAEs. This increase is dependent on the length of the set of nested dictionary sizes ($\mathcal{M}$). In our experiments with 5 nested SAEs, this increased training time with circa 50%.

While we conducted several ablations (Appendix H), many design choices specific to Matryoshka SAEs remain to be explored. For instance, explicitly enforcing the number of active latents for different latent groups within the Matryoshka SAE has not been studied and might offer even finer control over the granularity of learned features. Such variations could be promising avenues for future research.

Finally, our evaluation relies heavily on imperfect quantitative measures of feature quality and automated interpretability metrics and are mostly conducted a single LLM. For example, our evaluation of feature absorption relies primarily on first-letter classification tasks. While these standardized metrics provide valuable insights, they may not fully capture human-relevant aspects of feature interpretability or the practical utility of the learned representations for manual downstream analysis. Future work investigating the human interpretability of Matryoshka SAE latents and their applicability to practical interpretability tasks would be valuable.

## 6. Discussion and Conclusion

We have presented Matryoshka SAEs, a novel variant of sparse autoencoders that addresses key challenges with sparsity when scaling SAEs. By enforcing hierarchical feature learning through nested dictionaries, our approach achieves lower feature absorption rates and improved concept isolation compared to standard approaches. This demonstrates that the pathologies of scaling SAEs are not fundamental limitations, but rather artifacts of myopic training objectives that can be overcome through simple modifications to the training process.

Looking forward, the success of this approach suggests that many apparent problems with SAEs may have surprisingly simple solutions waiting to be discovered. As we work to understand increasingly large and capable neural networks, approaches to interpretability that scale as well will become essential to better understanding these systems.

## Impact Statement

This work advances mechanistic interpretability techniques for neural networks. Better understanding of model internals could help identify failure modes, verify intended behaviors, and guide the development of more reliable and safe AI systems. However, advances in model interpretability may also accelerate progress in large language model development and deployment, potentially amplifying both positive and negative societal impacts of these technologies.

## Acknowledgments

We are extremely grateful for feedback, advice, edits, helpful discussions, and support from Joel Becker, Gytis Daujotas, Julian D'Costa, Leo Gao, Collin Gray, Dan Hendrycks, Benjamin Hoffner-Brodsky, Mason Krug, Patrick Leaskm Hunter Lightman, Mark Lippmann, Charlie Rogers-Smith, Logan R. Smith, Glen Taggart, Adly Templeton, and Matthew Wearden.

This research was made possible by funding from Lightspeed Grants and AI Safety Support.

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

## A. SAE Training Details

We use the baseline SAE suite provided by SAE Bench (Karvonen et al., 2024a). It contains SAEs trained on layer 12 of Gemma-2-2B and layer 8 of Pythia-160M Biderman et al. (2023). The SAE Bench suite contains 6 proposed SAE variants.

| SAE Bench Variants |
|---|
| ReLU (Anthropic Interpretability Team, 2024) |
| TopK (Gao et al., 2024) |
| BatchTopK (Bussmann et al., 2024) |
| Gated (Rajamanoharan et al., 2024a) |
| JumpReLU (Rajamanoharan et al., 2024b) |
| P-Annealing (Karvonen et al., 2024c) |

*Table 1.* List of evaluated sparse autoencoder architectures

Our Matryoshka SAEs were trained in a directly comparable manner, including identical data ordering and hyperparameters.

## B. Code Implementation

This appendix shows an example code implementation to get the reconstruction and loss of a Matryoshka SAE. For the code implementation used in our experiments see `https://github.com/saprmarks/dictionary_learning/blob/main/dictionary_learning/trainers/matryoshka_batch_top_k.py`.

```python
def forward(self, input_data):
    # Compute features and apply sparsity
    features = relu((input_data - self.bias) @ self.encoder)
    sparse_features = batch_top_k(features)  # Same features as standard BatchTopK SAE

    # Dictionary size cutoffs
    dict_sizes = [2048, 6144, 14336, 30720, 65536]

    losses = []

    # Start with bias
    current_output = self.bias.clone()

    # Incrementally add reconstructions from each group
    for i in range(len(dict_sizes)):
        start_idx = 0 if i == 0 else dict_sizes[i-1]
        end_idx = dict_sizes[i]

        # Add contribution from this group of features
        group_features = sparse_features[:, start_idx:end_idx]
        group_weights = self.decoder[start_idx:end_idx]
        current_output = current_output + group_features @ group_weights

        # Store and compute loss
        losses.append(((current_output - input_data)**2).mean())

    # Sum all losses
    total_loss = sum(losses)

    return total_loss, current_output  # Return summed loss and final reconstruction
```

*Listing 1.* Example code implementation of Matryoshka SAE

## C. Toy Model Details

Here we provide implementation details for training the SAEs on our synthetic hierarchical feature dataset. Code can be found at `https://github.com/noanabeshima/matryoshka-saes`. The models are implemented in PyTorch

and use the following configuration:

**Model Architecture.** Both the Vanilla and Matryoshka SAEs use the following hyperparamaters:

- Input/output dimension: 20

- Number of latents: 20

- ReLU activation function

- Adaptive sparsity control targeting the ground-truth average $\ell_0$ of 1.2338

- Optimizer: Adam with learning rate 3e-2, betas=(0.5, 0.9375)

- Training steps: 40,000

- Batch size: 200

- Gradient clipping norm: 1.0

**Matryoshka-Specific Components.** Per batch, 10 prefix lengths are sampled from a truncated Pareto(0.5) distribution, with the full SAE always included. Specifically, each possible prefix length $\ell \in \{1, \ldots, m\}$, where $m$ is the total number of latents, is assigned a probability proportional to $P(\ell) \propto 1 - \left(\frac{\ell}{m}\right)^\alpha$ where $\alpha > 0$ controls how heavily the distribution favors shorter prefixes (we use $\alpha = 0.5$ in our experiments). This yields a monotonically decreasing probability distribution over prefix lengths. We then normalize this distribution and sample prefix lengths without replacement, always including the full prefix length $m$ to ensure the complete SAE is trained.

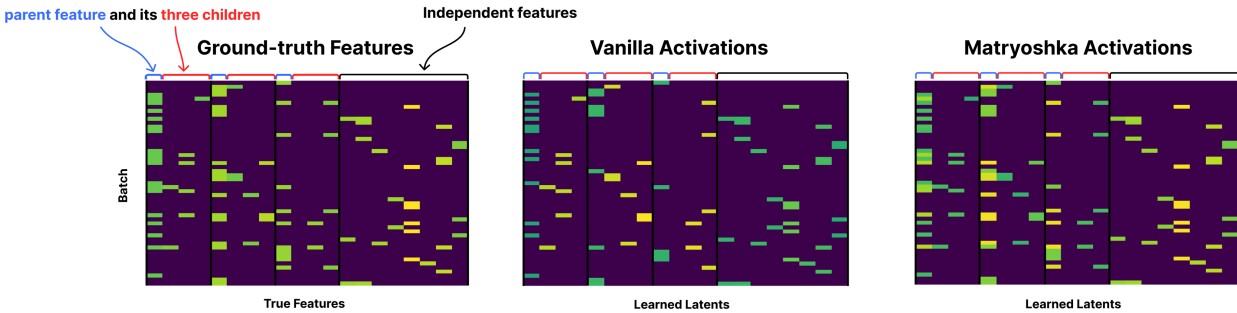

*Figure 11.* **Full feature absorption toy model activations** Ground-truth feature activations alongside Matryoshka SAE activations and Vanila activations. Notice that for the Vanilla SAE, parent latents (columns bracketed in blue) do not fire when their children (bracketed in red) fire.

# D. TinyStories examples

Figure 12 shows a set examples of features learned by the Matryoshka SAE trained on the TinyStories model. The first feature activates on female-related tokens, the second feature on adjectives indicating that something is small, whereas the third feature activates on nouns that refer to the location where a story takes place.

*Figure 12.* A set of example features learned by the Matryoshka SAE trained on the TinyStories model.

## E. Tree Methodology

To generate the hierarchical visualizations in sparselatents.com/tree_view, we implemented a variant of Masked Cosine Similarity (MCS), building upon the metric introduced in "Towards Monosemanticity" (Bricken et al., 2023).

### E.1. Modified Masked Cosine Similarity

The original MCS between two latents A and B is calculated as follows:

1. Compute the cosine similarity between activations of A and B, but only for tokens where latent A is active

2. Compute the cosine similarity between activations of A and B, but only for tokens where latent B is active

3. Take the larger of these two similarity values as the final MCS

Our approach modifies MCS in two ways:

1. We only consider the cosine similarity between A and B's activations on tokens where B is active, for a directed version of MCS.

2. We scale this directed similarity by the ratio $\max(\text{B activations})/\max(\text{A activations})$, which we hoped would down-weight relationships involving latents with minimal activation

We use a more conservative similarity threshold of 0.6 compared to the original implementation.

### E.2. Tree Construction Process

The tree generation proceeds through these steps:

1. Starting with a parent latent (e.g., S/1/12), we identify all latents in a wider SAE model (e.g., S/2) that exceed our directed MCS threshold

2. For each identified S/2 latent, we recursively apply the same methodology to find children in S/3

3. We continue this process to build the complete hierarchical structure

### E.3. Handling Multi-Parent Relationships

The resulting structure often forms a directed acyclic graph (DAG) rather than a strict tree, as some latents in deeper layers (e.g., S/3) may have relationships exceeding the similarity threshold with multiple parent latents in the previous layer (e.g., S/2).

To simplify visualization, we assign each multi-parent latent exclusively to the parent with which it exhibits the highest similarity score. While this approach necessarily obscures some of the complexity in the underlying network structure, it enables clearer visualization and interpretation.

### E.4. Interpretation

It's important to note that these tree visualizations should not be interpreted as comprehensive representations of the full model structure. Rather, they serve as targeted explorations highlighting sets of latents that potentially demonstrate feature absorption relationships when examined together. They provide a window into how features may be organized and recomposed across different model widths.

## F. Investigating the sub-SAEs

In addition to using the complete Matryoshka SAEs, we also investigate using only a number of the sub-SAEs. For this analysis, we train a Matryoshka SAE with 36864 latents and 5 subgroups of {2304, 4608, 9216, 18432, and 36864}. For fair comparison, we train five standard BatchTopK SAEs with dictionary sizes matching our nested sub-SAEs (2304, 4608, 9216, 18432, and 36864), each calibrated to match the effective sparsity of the corresponding sub-SAE (average L0 norms of 22, 25, 27, 29, and 32 respectively).

In Figure 13, we find a distinct activation patterns across different latent groups, with early latents showing consistently higher activation rates due to their participation in multiple reconstruction objectives.

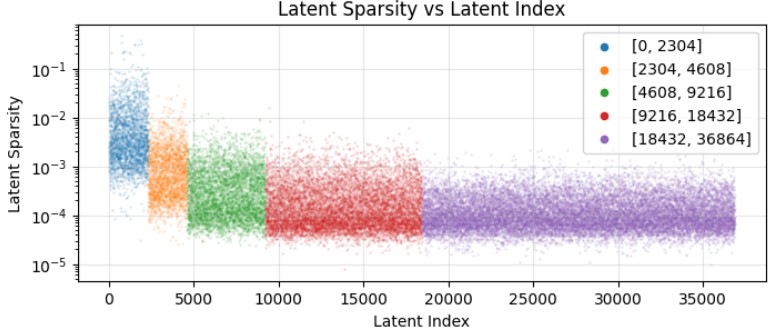

*Figure 13.* **Latent frequency per latent index.** Distribution of latent activations across different groups in the Matryoshka SAE. Each color represents a different nested dictionary size. Early latents (blue) show higher average activation rates, reflecting their use in multiple reconstruction objectives.

By using only a limited number of the sub-SAEs of the single Matryoshka SAE, we can construct dictionaries of different sizes. Figure 14 shows the reconstruction performance of the sub-SAEs compared to their matched BatchTopK SAEs. Although the BatchTopK SAEs explain a larger percentage of the variance of the input, the difference in degradation of downstream language model cross-entropy loss decreases when more sub-SAEs are used.

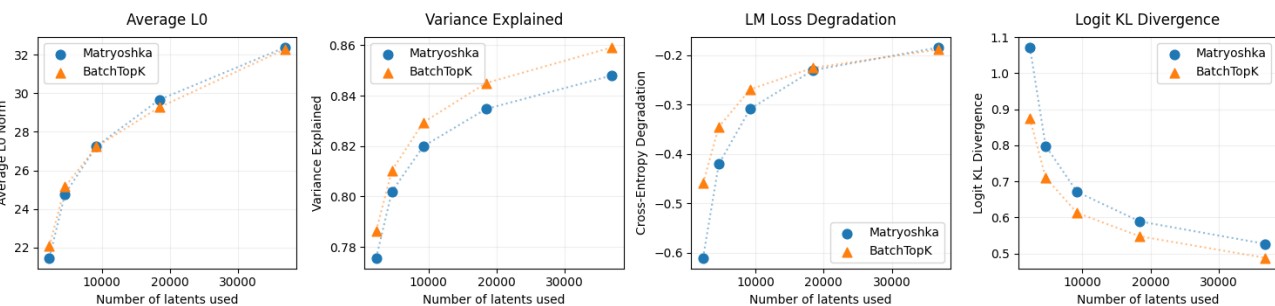

*Figure 14.* **Reconstruction performance of sub-SAE.** Although the BatchTopK SAEs explain a larger percentage of the variance of the input, the difference in degradation of downstream language model cross-entropy loss decreases when more sub-SAEs are used.

### F.1. Hierarchies of Matryoshka Features

The nested training objective of Matryoshka SAEs is designed to encourage the formation of a feature hierarchy. Early latents, belonging to smaller sub-SAEs, are pressured to capture broad, general concepts since they must reconstruct the input independently. Subsequent latents in larger sub-SAEs can then learn more specific features, potentially representing refinements or components of the earlier, more general features, without the same incentive to absorb or fragment these foundational concepts.

We can visualize these learned hierarchies by examining the co-occurance between features. Figure 15 illustrates such a hierarchy. In this hierarchy, we observe a clear parent-child relationship structure. The root latent (103) represents a general "sports team names" concept that activates broadly on any mention of sports teams. This general feature branches into more specific subcategories, such as rankings of sports teams and news headlines containing sports teams.

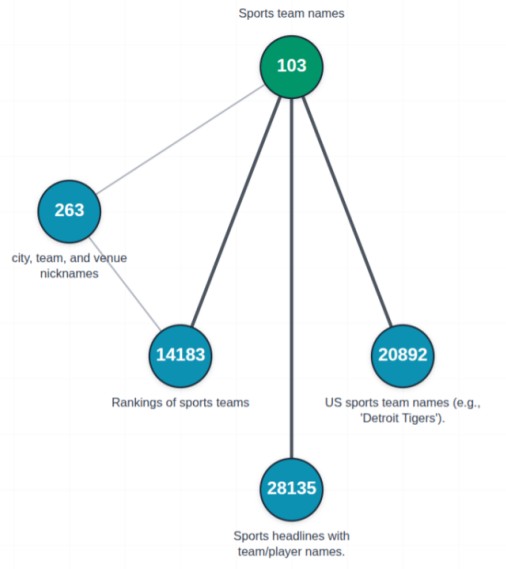

*Figure 15.* Hierarchy of features related to sports names. The thickness of the connections indicates the fraction of inputs on which the child features co-occur with their parent features.

### F.2. Investigating Composition with Meta-SAEs

We train meta-SAEs on the decoder matrices of both architectures. Each meta-SAE has a dictionary size one-quarter that of its input SAE and uses an average of 4 active latents. The variance explained by these meta-SAEs serves as a proxy for shared information between latents.

Meta-SAEs explain substantially more variance in BatchTopK SAE decoder directions compared to Matryoshka SAE directions (Figure 16). For the largest dictionary size Meta-SAEs explain 55% of variance in BatchTopK decoder directions, whereas they only explain 42% of variance in Matryoshka SAE decoder directions.

This indicates that Matryoshka SAE latents are more disentangled, with less shared information between them. The effect strengthens with dictionary size: BatchTopK SAEs show increasing levels of shared structure while Matryoshka SAEs maintain relatively constant levels of disentanglement.

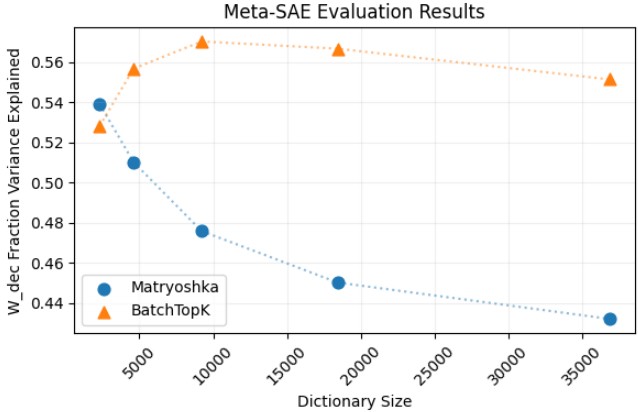

*Figure 16.* Meta-SAE Evaluation Comparison between Matryoshka SAEs and standard BatchTopK SAEs.

## G. Evaluations with Board Game Models

Karvonen et al. (2024c) evaluated SAEs on specialized models trained exclusively on chess and Othello games, leveraging the well-defined ground truth features in these domains. Their evaluation created two key metrics:

- Board reconstruction: The ability to reconstruct the game board from the SAE latents

- Feature coverage: The alignment of SAE latents with predefined board game features

While these toy models offer a controlled environment for testing, we find that the results do not show significant differentiation between Matryoshka SAEs and other evaluated architectures (ReLU, P-Anneal, Gated, and TopK) in terms of these metrics. In the low L0 regime (L0 ≤ 150), Matryoshka performs comparably to other architectures on both metrics. Notably, the best overall performance achieved by each architecture tends to occur in this lower L0 range, with peak scores being fairly comparable across architectures.

However, Matryoshka shows some degradation in performance at higher L0 values. This behavior may be partially explained by the model architecture: Chess-GPT uses a relatively small hidden dimension of 512, meaning that high L0 values (≥ 150) represent a significant fraction of the model's representational capacity. This differs substantially from our main experiments on Gemma-2B, where even our largest L0 values represent a much smaller fraction of the model's hidden dimension.

Given that our primary interest lies in understanding SAE behavior on large-scale language models that better reflect real-world applications, we focus our main analysis on results from Gemma-2-2B. However, we include these toy model results for completeness and to facilitate comparison with prior work. Figure 17 shows the performance comparison on the two metrics from the chess model evaluation.

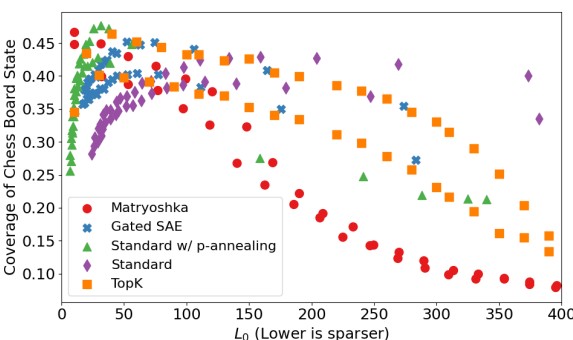 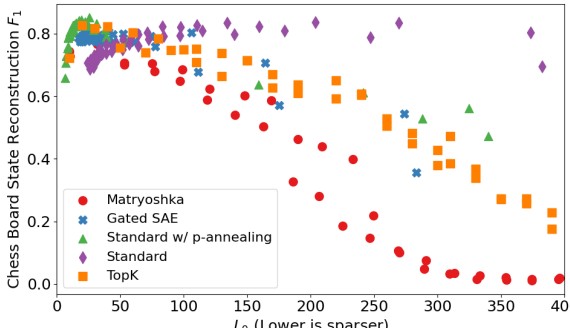

*Figure 17.* Evaluation results on the ChessGPT toy model showing (left) feature coverage and (right) board reconstruction accuracy across different architectures. While Matryoshka SAEs show comparable performance at low L0 values, they slightly underperform at higher L0s compared to other architectures.

## H. Ablations

To better understand the key components of Matryoshka SAEs and validate our design choices, we conduct a series of ablation studies. These experiments systematically modify different aspects of the architecture while keeping other variables constant. All ablation studies were trained on 200M tokens from The Pile (Gao et al., 2020), with evaluations performed on layer 12 of Gemma-2-2B. We examine both 16k and 65k dictionary sizes and find differences become more pronounced at larger sizes, so we focus on 65k in our ablation studies.

### H.1. Ablation: Loss Weighting

The standard Matryoshka SAE equally weights the reconstruction loss for each nested dictionary size. However, we can interpolate between Matryoshka and standard BatchTopK behavior by weighting the losses according to dictionary size. The BatchTopK SAE can be thought of as a special case of Matryoshka with a single group.

We implement a Weighted Matryoshka variant that assigns loss weights proportional to the number of new latents in each group. For nested dictionaries using [25%, 50%, 100%] of total latents, this yields weights [0.25, 0.25, 0.5] compared to the standard [0.33, 0.33, 0.33]. This weighting scheme emphasizes accurate reconstruction of the full dictionary while maintaining some pressure for hierarchical feature learning.

With our ablation training budget of 200M tokens, the Weighted Matryoshka achieves downstream cross-entropy loss similar to BatchTopK SAEs, outperforming the equally-weighted variant. However, this comes at the cost of reduced feature quality - the weighted variant performs worse on feature absorption and spurious correlation removal metrics, suggesting that equal weighting is important for maintaining the hierarchical benefits of Matryoshka SAEs. Given that the cross-entropy gap between standard Matryoshka and BatchTopK disappears with larger training budgets, larger L0s, and larger dictionary sizes (as shown in Section 4.3), these results support using equal weighting to preserve feature quality without significant reconstruction trade-offs.

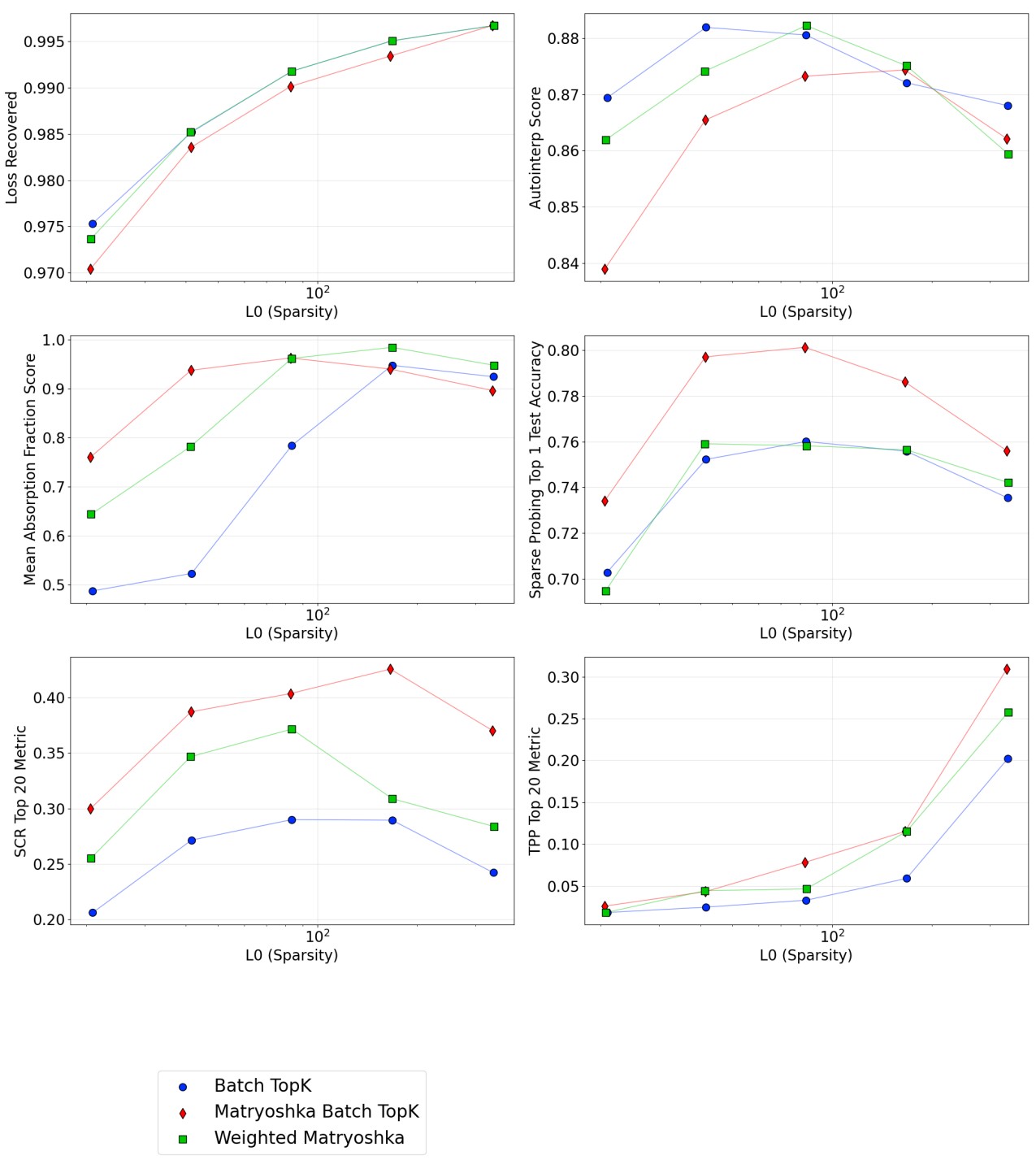

*Figure 18.* Evaluation Results comparing the Matryoshka, Weighted Matryoshka, and BatchTopK 65K width Gemma-2-2B SAEs.

## H.2. Ablation: Stop Gradients

We investigate how gradient flow between different dictionary sizes affects the learning dynamics of Matryoshka SAEs. In the standard implementation, the reconstruction for each nested dictionary incorporates the reconstructions from smaller dictionaries. This allows gradients to flow through the entire chain of reconstructions - when optimizing the loss for larger dictionaries, gradients can affect how smaller dictionaries learn their features. In our stop gradient variant, we detach each partial reconstruction before it's used in the next one, forcing each group of latents to learn independently of the others.

These changes effectively isolate the training of each nested dictionary, preventing larger dictionaries from influencing how smaller ones learn features. Our evaluation shows mixed results: while the stop gradient variant achieves better feature absorption scores at low L0 values, it shows degraded performance on loss recovered, sparse probing, and Spurious Correlation Removal (Figure H.2). Given these degradations, we did not pursue this variant further, though future work with more sophisticated metrics or qualitative analysis may reveal additional benefits to this approach.

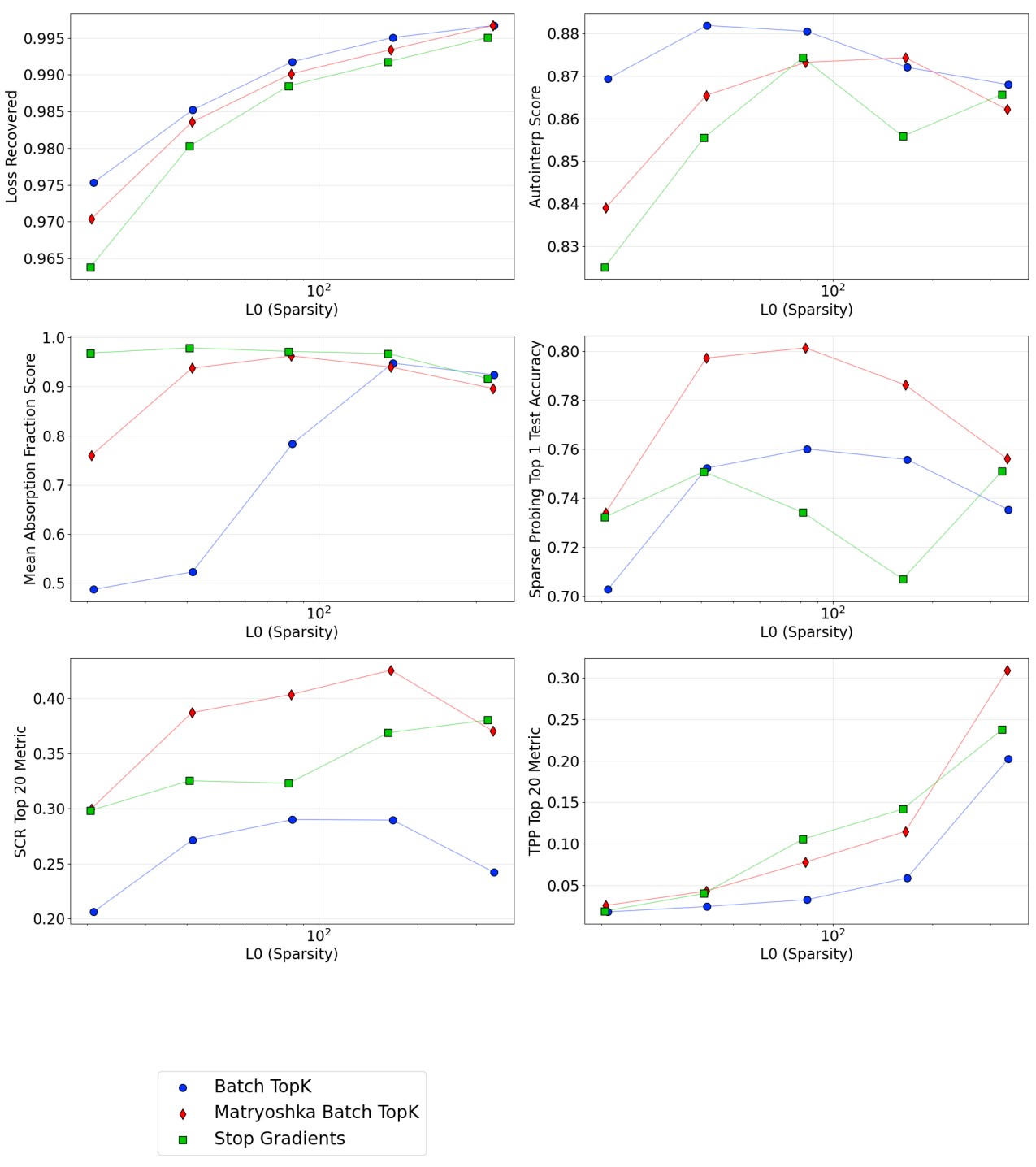

*Figure 19.* Evaluation Results comparing the Matryoshka, Stop Gradient Matryoshka, and BatchTopK 65K width Gemma-2-2B SAEs.

## H.3. Ablation: Number of Nested Dictionaries

We investigate how the number of nested dictionaries affects Matryoshka SAE performance. Our standard implementation uses five nested dictionaries, with each group representing the following fractions of the total latents: [1/32, 1/16, 1/8, 1/4, (1/2 + 1/32)]. We compare this against two variants:

- A three-group variant with ratios [1/8, 1/4, (5/8)]

- A ten-group variant with ratios (1/16384)[96, 152, 241, 383, 607, 964, 1531, 2430, 3857, 6123], derived from logarithmically spaced values between 128 and 8192 and normalized to sum to the total dictionary size. We chose this logarithmic spacing to avoid the extremely small dictionary sizes that would result from extending our standard geometric progression to 10 groups.

The three-group variant improves on loss recovered compared to our standard five-group implementation but performs worse on feature absorption and SCR, suggesting it interpolates between Matryoshka and BatchTopK behavior. Our ten-group variant slightly improves on feature absorption but performs significantly worse on loss recovered and automated interpretability metrics. Based on these results, we find that while the exact number of groups is not critical, five groups provides a reasonable balance between the trade-offs present in these metrics (Figure H.3).

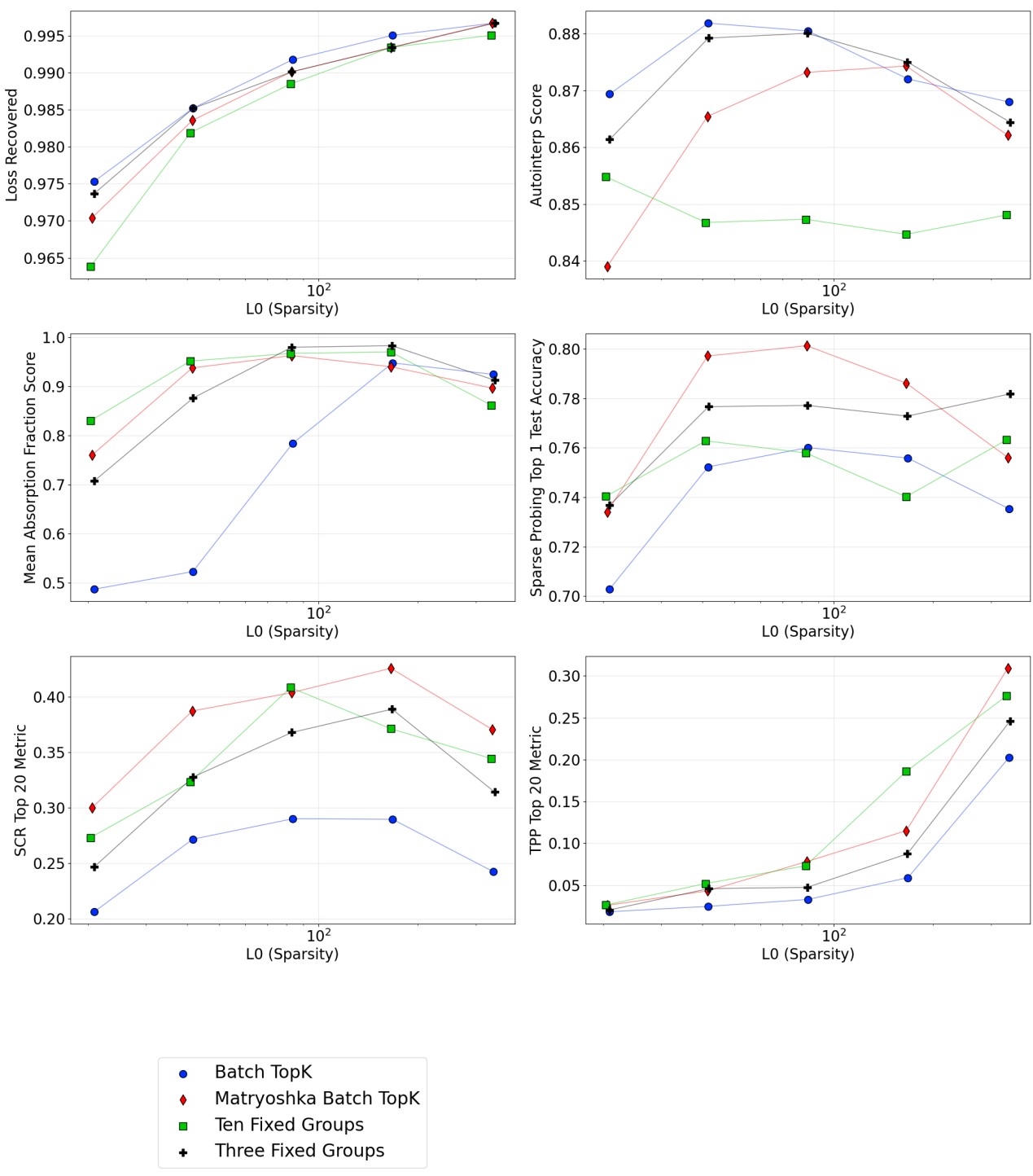

*Figure 20.* Evaluation Results comparing Matryoshka variants with different numbers of nested dictionaries (3, 5, and 10 groups) against BatchTopK SAE. All models use 65K width on Gemma-2-2B

## H.4. Ablation: Randomly Sampled Dictionary Sizes

As an alternative to fixed dictionary sizes, we explored dynamically sampling prefix lengths from a truncated Pareto distribution during training. Specifically, each possible prefix length $\ell \in \{1, \ldots, m\}$, where $m$ is the total number of latents, is assigned a probability proportional to $P(\ell) \propto 1 - \left(\frac{\ell}{m}\right)^\alpha$ where $\alpha > 0$ controls how heavily the distribution favors shorter prefixes (we use $\alpha = 0.5$ in our experiments). This yields a monotonically decreasing probability distribution over prefix lengths. We then normalize this distribution and sample prefix lengths without replacement, always including the full prefix length $m$ to ensure the complete SAE is trained.

Random sampling could theoretically create a more continuous feature hierarchy by exposing the model to diverse dictionary sizes throughout training. We evaluated both strategies on 16k-width SAEs with three different configurations for the number of sampled prefixes (3, 5, and 10 groups).

On our current evaluation metrics, random sampling showed similar performance to fixed dictionary sizes, with minor degradation in spurious correlation removal and sparse probing performance, as shown in Figure 21. Random sampling may also introduce additional complexity for distributed training across multiple GPUs. While our evaluation framework may not capture all relevant aspects of feature quality, we opted for fixed dictionary sizes for our main experiment given these considerations.

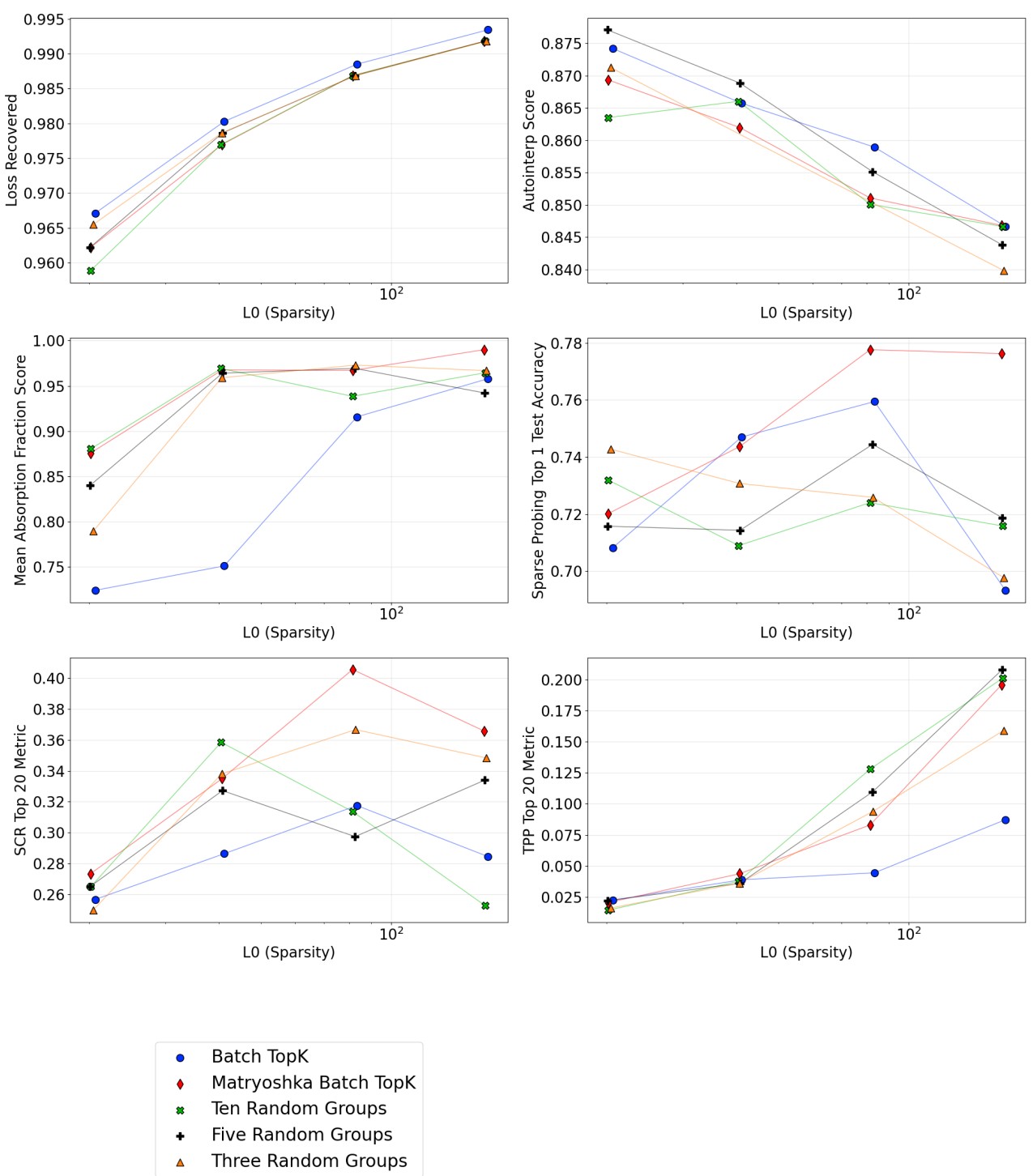

*Figure 21.* **Random vs Fixed Dictionary Sizes.** Comparison of Matryoshka SAE variants using randomly sampled dictionary sizes against fixed dictionary sizes and BatchTopK baseline. We evaluate configurations with 3, 5, and 10 groups on 16k-width SAEs trained on Gemma-2-2B. Random sampling shows similar or slightly worse performance compared to fixed dictionary sizes across most metrics.

