# OpenReview forum: "Learning Multi-Level Features with Matryoshka Sparse Autoencoders"
_ICML.cc/2025/Conference — ICML 2025 poster_

### Official Review · Reviewer_3MV4 · 2025-03-11

**Overall Recommendation:** 4

**Summary:**

This paper introduces a simple but novel approach for training SAEs with a nested structure in the feature space. As a consequence of this training, the authors present results suggesting that Matryoshka SAEs are more adept at overcoming feature splitting and feature absorption issues currently facing SAEs. The nested feature space concept of Matryoshka SAE is inspired by a previous work which introduces Matryoshka representations.

**Claims And Evidence:**

The claims made in the submission seem well supported by the experiments.

**Essential References Not Discussed:**

None

**Experimental Designs Or Analyses:**

Overall, the experimental design and analysis is sound.

**Methods And Evaluation Criteria:**

The proposed methods and evaluation criteria are sound.

**Other Comments Or Suggestions:**

None

**Other Strengths And Weaknesses:**

Strengths:
- Paper is well written and has clear language
- The set of experiments presented seems relevant and extensive. This includes presenting results for which their method is not necessarily the best (e.g., reconstruction and game board experiments)

**Questions For Authors:**

Overall, the paper looks good, but I do have the following inquiry about the experiments: The extent of the large-scale experiments is on Gemma-2B. I understand that restrictions on compute may prohibit running experiments on additional models of similar or larger size, but evaluation on a single, relatively small language model is a limitation itself. I'm wondering if the authors have any insight on how the results may change as a function of model scale, architecture, or pretraining data?

**Relation To Broader Scientific Literature:**

This work connects well with the existing literature. Feature splitting and feature absorption are known limitations of SAEs; this work presents a method which overcomes these issues while drawing inspiration from previous works, namely the SAE and Matryoshka representation learning literature.

**Theoretical Claims:**

N/A

---

> ### Author Rebuttal · Authors · 2025-03-31
>
> Thank you for your positive review of our paper and thoughtful feedback! We are grateful for your recognition that our work is well-written with clear language, presents relevant and extensive experiments, and effectively addresses known limitations of SAEs.
>
> Regarding different model scales and architectures: While computational constraints limited testing on a wider range of the largest models, we have reasons to believe the benefits of Matryoshka SAEs will generalize and potentially even increase with model scale. Feature absorption and splitting are fundamental challenges in SAEs rather than model-specific issues. Based on theoretical considerations, larger models with more complex feature hierarchies might actually benefit more from our approach, as they contain richer hierarchical structures that Matryoshka SAEs are designed to preserve. Different architectures might show varying degrees of improvement, with models having stronger hierarchical representations potentially showing more pronounced benefits. We acknowledge the current experimental scope as a limitation and will add a note to this effect in the camera-ready version.
>
> Thank you again for your positive assessment and valuable question. We hope our response reinforces your support of our paper.

---

### Official Review · Reviewer_phSQ · 2025-03-11

**Overall Recommendation:** 5

**Summary:**

The authors suggest a novel training objective for sparse autoencoders to address the issues of feature splitting, feature absorbtion and feature composition.
They test this idea on a toy, synthetic dataset explicitly designed to demonstrate improvements, then a 4-layer transformer-based language model traied on TinyStories, then Gemma-2-2B checkpoints (open-source language model from Google).
They demonstrate strong improvements through a variety of experiments, assuming well-tuned baselines.

**Claims And Evidence:**

The authors claim that despite worse reconstruction performance, Matryoshka SAEs better achieve the original goals of SAEs.
These claims are well-supported by their experiments in Section 4.3.

**Essential References Not Discussed:**

There are no essential references not discussed.

**Experimental Designs Or Analyses:**

The experimental design is excellent.
Baselines appear to be well-tuned.
I did not find any issues in the soundness or validity of the experimental designs.

**Methods And Evaluation Criteria:**

The methods (synthetic data, TinyStories, and Gemma-2-2B) and evalution criteria all make sense.
Ideally there would be human evaluation as well, but the authors mention this as a limitation and have a public SAE latent viewer for TinyStories.
I am very satisfied with the evaluation.

It would be interesting to see the effects of steering with the different latent subgroups.
But I'm happy for that to be future work.

**Other Comments Or Suggestions:**

* There's a fair amount of whitespace. Can you use that whitespace to make bigger (taller) figures?

**Other Strengths And Weaknesses:**

The simplicity of the idea is phenomenal.
Most SAE modifications are in the activation layer.
It's exciting to see novelty in the training objective, and I would love to see how this objective composes with other SOTA activation layers like JumpReLU.

**Questions For Authors:**

* Can you explain why the PCA baseline has the highest variance explained and the lowest CE loss (figure 5)? Is it because PCA doesn't have a sparsity constraint?

**Relation To Broader Scientific Literature:**

This work fits in with other novel SAE architecture/objectives like TopK, JumpReLU, etc.
It explains its position within the broader literature well.

**Theoretical Claims:**

There appear to be no theoretical claims.

---

> ### Author Rebuttal · Authors · 2025-03-31
>
> Thank you for your very positive review of our paper! We were happy to read your comment about the experimental design being excellent and your appreciation for the simplicity and novelty of the idea.
>
> Furthermore, we appreciate your suggestion to include a few feature examples, which we will do in the camera-ready version. We will also make sure to minimize the whitespace in this final version. Regarding the PCA baseline: Your intuition is exactly correct. The high variance explained and low CE loss are indeed due to the lack of a sparsity constraint, allowing it to use its full capacity for reconstruction unlike the SAEs. We will add a sentence clarifying this in the camera-ready version.
>
> Thank you again for your strong support and enthusiastic review!

---

> > ### Comment · Reviewer_phSQ · 2025-04-04
> >
> > Thank you for the clarification.

---

### Official Review · Reviewer_HNXv · 2025-03-11

**Overall Recommendation:** 4

**Summary:**

This paper presents Matryoshka SAEs, inspired by Matryoshka representation learning, that learns a nested series of SAEs simultaneously to address issues such as feature splitting and feature absorption. Comparisons with multiple well-established baseline SAEs demonstrate Matryoshka SAEs’ superior quality to overcome these problems as well as exhibiting better concept isolation performance. Though their reconstruction performance is worse than baselines, Matryoshka SAEs show better automatic interpretability metrics and better scalability.

**Claims And Evidence:**

Yes.

**Essential References Not Discussed:**

I have not found essential missing references.

**Experimental Designs Or Analyses:**

Yes. All designs are valid, although I have the following concerns related to Section 4.1, Toy Model Demonstration of Feature Absorption.
- Here, different from real-world scenarios, the authors are not training sparse or overcomplete autoencoders; instead, the number of latents is set equal to the ground-truth. According to the superposition hypothesis (e.g., [1]), an SAE is trained to approximate the high-dimensional hypothetical disentangled model using low-dimensional features from the observed model, which translates into d < L in this setup. Will the Matryoshka SAE maintain its superior performance under this condition?
- As detailed in Appendix E, there are two important Matryoshka-specific components. First is an “adaptive sparsity control targeting the ground-truth average l_0 of 1.2338”. Could the author be more specific on what this means? How is this ground-truth information incorporated into the training process? The second component is the “l_1 sparsity penalty on normalized activations”. Does this imply that the vanilla SAE lacks an l_1 penalty? If so, then it would be beneficial to include ablations on the vanilla SAEs that also incorporate these biases.
- In this setup, all ground-truth features are orthogonal to each other; however, in a more realistic setting, one would expect (at least) the parent feature and a child feature to naturally have a higher cosine similarity score compared to inter-group cosine similarity. I have this concern because I’ve spotted the correlation between the parent feature and the children features in the Matryoshka SAE as depicted in Figure 3, which is unexpected given the goal of avoiding absorption. Will this effect be amplified in a non-orthogonal setup?

[1] Toy Models of Superposition. Section link: https://transformer-circuits.pub/2022/toy_model/index.html#motivation-superposition

**Methods And Evaluation Criteria:**

Yes. The Matryoshka structure is well-suited to address the feature splitting/absorption problems, and the evaluation metrics used as well as the choice of baseline SAEs all follow the community standard.

**Other Comments Or Suggestions:**

I would suggest the authors mention the ablation studies in Appendix B (for example, use one sentence) into the main text, especially regarding loss weighting. It is noteworthy that average weighting works the best in this context.

**Other Strengths And Weaknesses:**

The paper is clearly written. Incorporating the Matryoshka learning idea into SAE is novel. It shows the potential to address problems in large-scale SAEs trained on LLMs.

**Questions For Authors:**

I have the following questions for authors.
- In the first-letter experiment, you first discover a direction via probing, and then “find the corresponding latents and measure when these SAE latents fail to activate on tokens starting with that letter and get absorbed into specific token-aligned latents.” How is this finding process performed? Is it through cosine similarity comparison? If this is the case, have you checked the token distribution with a specific letter, to avoid potential issues that lead to identifying a local feature/token feature with “holes” due to an imbalanced distribution (for example, corpus starts with L dominated by “Lily”)?
- In the sparse probing experiment, the performance drops at higher sparsity, both in Figure 7 and Figure 10. This looks like a warning sign that when increasing the dictionary size, the Matryoshka SAE loses the ability (which it has when scaling is not performed) to some extent to isolate the target concept. Do you have an explanation or discussion on this phenomenon?
- I wonder how the batch size setup in BatchTopK would affect the result. I suspect you chose 2048 to maintain consistency to the baselines, but if a stability result on this hyperparameter can be provided it would be beneficial.

**Relation To Broader Scientific Literature:**

The proposed idea of incorporating Matryoshka representation learning into SAEs helps address several crucial problems in the current SAE community, which improves interpretability of the family of sparse autoencoders.

**Theoretical Claims:**

N/A

---

> ### Author Rebuttal · Authors · 2025-03-31
>
> Thank you for your positive review and insightful questions. We appreciate your assessment that our paper is clearly written and the idea is novel. We address your key questions below:
>
>
> **Regarding the toy model demonstration:**
>
> 1. **On d < L setup:** You correctly note the toy model uses a non-overcomplete set-up, for simplicity and illustrative purposes. The toy model is not meant to be a realistic depiction of features in LLMs, but rather a simplified demonstration to illustrate a specific scenario where vanilla SAEs struggle with feature absorption while Matryoshka SAEs excel. It provides a controlled environment to highlight the core mechanism of our approach. We chose to use a small number of latents to make it easier to visualize and show the resulting learned features, but expect that similar results would hold for toy models with more ground-truth features.
>
> 2. **On adaptive l1 penalty:** Thanks for pointing this out, we should have been clearer here. The adaptive sparsity control is not a Matryoshka SAE-specific component, but rather a component we use in both SAEs. The only difference between the two SAEs is the nested reconstruction objective of the Matryoshka SAE. We will update this in the camera-ready version.
>
> 3. **On orthogonal features:** You raise an important point about the correlation between parent and child features. Although the parent and child features are orthogonal, the resulting combined activations of the "parent + child" are not orthogonal to each other, due to the shared parent component. This is why we observe a small correlation between parent and child features in the Matryoshka SAE (Figure 3), as the disentanglement is, although close, not perfect. Given that there is some earlier work by Park et al [1] that suggests that in real-world settings, parent and child features are often orthogonal, we think it is reasonable to use orthogonal features in our toy model as well.
>
> **Regarding the first-letter experiment:**
> To find the relevant latents for the first-letter task, we use a train and test set of single token inputs sampled directly from the tokenizer, unweighted by token frequency. We first train logistic regression probes on residual stream activations to define the ground truth direction for each letter. We then use k-sparse probing on the SAE latents to identify the main latent(s) representing that feature. Specifically, we start with k=1 to find the primary latent, then incrementally increase k, adding additional latents to our "main latents" set when they improve the F1 score by more than a threshold (τ=0.03). For each token, we then measure when these main SAE latents fail to fully activate on tokens starting with that letter (their projection onto the ground truth direction is less than the model's projection) while other latents (the absorbing latents) compensate on these tokens. Since we use the SAEBench implementation for our evaluation, we refer to their paper for more details [2].
>
> **Regarding sparse probing performance:**
> The performance drop at higher sparsity in Figure 7 is indeed interesting, but it's important to note that this phenomenon occurs across many other baseline architectures as well, not just in Matryoshka SAEs. This appears to be a general characteristic of SAEs rather than a limitation specific to our approach. We want to note that the performance drop in sparse probing when scaling up the dictionary size from 16k to 65k (Figure 10) is very minor (0.775 -> 0.772) and we don't believe this is a significant effect.
>
> **Regarding BatchTopK batch size:**
> We chose a batch size of 2048 to maintain consistency with baselines as you suspected. While we didn't perform extensive ablations on this hyperparameter, we suspect that for reasonably large batch sizes it is not a critical hyperparameter, and that standard BatchTopK SAEs and Matryoshka BatchTopK SAEs will be similarly affected by changes in batch size.
>
> We hope these explanations address your questions. We would appreciate your consideration if these clarifications strengthen your support of the paper.
>
> [1] Park, Kiho, et al. "The geometry of categorical and hierarchical concepts in large language models." arXiv preprint arXiv:2406.01506 (2024).
>
> [2] https://www.neuronpedia.org/sae-bench/info#feature-absorption

---

### Official Review · Reviewer_gKza · 2025-03-23

**Overall Recommendation:** 4

**Summary:**

This paper aims to improve concept learning in sparse autoencoders (SAEs), which are models that have recently become popular as a means to disentangle features from large deep models, particularly LLMs, into a sparse set of disentangled concepts. This work focuses on the problem with existing SAEs of choosing the optimal size of the latent dimension, where large sizes may help capture more concepts, but cause them to have irregular and inconsistent granularities. Specifically, it looks into the problems of feature absorption, feature splitting, and feature composition. It proposes an alternative training objective inspired by Matryoshka representation learning, where the SAE is forced to reconstruct the input feature independently using varying latent sizes, enforcing some latents to encode coarse concepts and others to encode more specialized, fine-grained concepts. Experimental evaluation is performed to show that the proposed Matryoshka SAEs are performant and interpretable, while also avoiding the aforementioned problems with existing SAEs.

## Update after rebuttal
Thank you for your detailed response. I agree that this paper should be accepted.

**Claims And Evidence:**

The claims are generally supported by evidence. Some of the experimental evaluation appears to be limited to specific tasks (e.g. first letter recognition) as discussed in the Weaknesses section below, and it would help to expand the evaluation to more tasks.

**Essential References Not Discussed:**

None that I am aware of.

**Experimental Designs Or Analyses:**

The experimental design and analyses appear to be sound. It would help to have a more diverse set of experiments however, as discussed in the Weaknesses section below.

**Methods And Evaluation Criteria:**

Yes, they make sense.

**Other Comments Or Suggestions:**

None.

**Other Strengths And Weaknesses:**

## Strengths
1. This work deals with an important problem—while many attempts have been made to Pareto-improve the reconstruction error and sparsity tradeoff, this work focuses on improving the interpretability of SAEs and addresses issues caused by scale, which is a valuable contribution.
2. Compared to the baselines, Matryoshka SAE performance improves with scale, which is promising.
3. The idea of learning concepts at different granularities makes intuitive sense and the proposed approach generally appears to be sound.
4. The paper is well written and easy to follow.

## Weaknesses
1. The training objective of the Matryoshka SAE as per Equation 5 only enforces that latent subsets from $0:m$ for all $m\in\mathcal M$ must independently be able to reconstruct the features. In particular, given $m_1<m_2$, there is no constraint that any latent outside of the first $m_1$ latents but within the first $m_2$ latents must activate at all. As a result, why should the SAE learn any specialized concepts at all? Is it simply that learning such concepts helps performance? An analysis of this, with respect to SAE size and reconstruction error, would be interesting to have, both in the toy setup and with real data.
2. In addition to the previous point, it might be interesting to see if it helps to enforce specific values of $K$ for different latent granularities, to control how many coarse and fine-grained concepts the model learns.
3. Currently there doesn't seem to be any evaluation to check how parent and child concepts relate, except in the toy setup. It would be useful to see if similar parent-child trees could also be constructed in the more realistic setups used in Section 4.2 and 4.3.
4. Generally, the evaluations in Section 4.2 and 4.3 are observation-based and limited to very specific tasks, such as first letter recognition. While this is promising, it would be helpful to also extend to at least a few more tasks, to assess the generality of the method.
5. The "truncated Pareto distribution" described in L210 is unclear, and there is no cited reference. A clarification would be helpful.
6. In the experiments in Section 4.2, it would help to also show qualitative results with Matroyshka SAEs. L284-291 claims it avoids feature absorption, but without a qualitative figure. Additionally, what happens if $K$ is varied in this experiment?
7. The metrics used in Figure 6 need to be defined precisely, they are currently unclear.

**Questions For Authors:**

Please refer to the Weaknesses section above. Overall I believe this work provides a valuable contribution and should be accepted. However, it would be helpful to have clarifications on the concerns raised in the rebuttal.

**Relation To Broader Scientific Literature:**

This work builds upon existing literature that develops SAEs as a mechanistic interpretability tool to disentangle concepts learnt by deep models such as LLMs. While most prior research on improving SAEs has focused on improving sparsity of SAEs (e.g. TopK SAE, Gao et al. 2024), this work looks at improving the interpretability of SAEs, and to avoid the recently reported problems of feature absorption, splitting, and composition.

**Theoretical Claims:**

No theoretical claims.

---

> ### Author Rebuttal · Authors · 2025-03-31
>
> Thank you for your detailed feedback and valuable suggestions. We are particularly encouraged that you found the paper well-written, addressing an important problem, and believe it is a valuable contribution that should be accepted.
>
> We address your key questions below:
>
> 1. You asked why specialized concepts emerge without explicit constraints. As you suggest, the driving force is indeed the reconstruction loss. The overall objective (summed over all dictionary sizes) incentivizes the full SAE to minimize reconstruction error. Specialized latents emerge naturally because they capture finer details present in some inputs that cannot be adequately represented by the limited number of coarser features alone, thus minimizing the total reconstruction error across the dataset. Our experiments (Appendix C) confirm that later latents, while activating less frequently, significantly contribute to reconstruction quality.
>
> 2. We appreciate this suggestion! Explicitly enforcing the number of active latents per group might indeed further improve granularity control. We'll include this in our discussion as a promising future research direction.
>
> 3. Thanks for the suggestion! We agree that it would be beneficial to include qualitative examples of parent-child relationships found in Matryoshka SAEs. We will include these in the camera-ready version.
>
> 4. You noted the absorption metric relies on the first-letter task. This is correct and follows the methodology established by SAE Bench [1] and Chanin et al. [2], benchmarks widely used in the community. While we acknowledge that broadening the tasks for measuring absorption could be beneficial (and will note this in our limitations), developing improved absorption metrics is beyond the scope of this work. Crucially, however, we want to emphasize that our other key evaluations, such as sparse probing and concept isolation, do utilize a diverse range of classification tasks.
>
> 5. Thanks for pointing out this omission. We sample the prefix lengths from a discrete distribution inspired by the Pareto distribution. Specifically, each possible prefix length $l \in \{1, \ldots, m\}$, where $m$ is the total number of latents, is assigned a probability proportional to:
> $$
> P(l) \propto 1 - \left( \frac{l}{m} \right)^{\alpha}
> $$
> where $\alpha > 0$ controls how heavily the distribution favors shorter prefixes (we use $\alpha = 0.5$ in our experiments). This yields a monotonically decreasing probability distribution over prefix lengths. We then normalize this distribution and sample prefix lengths without replacement, always including the full prefix length $m$ to ensure the complete SAE is trained. We will add this to the camera-ready version.
>
> 6. Figure 4 shows absorption in vanilla SAEs. As requested, we will add a corresponding figure for the Matryoshka SAE in the camera-ready version, demonstrating how the same feature (e.g., "female tokens") remains distinct (see https://sparselatents.com/tree_view?loc=residual+pre-attn&layer=3&sae_type=S%2F2&latent=65 for the example). We did not experiment with varying $K$ in this experiment, but we extensively tested this quantitatively in section 4.3.
>
> 7. Thank you for pointing out that the metrics in Figure 6 are not clearly defined. The metrics in Figure 6 are calculated the same as in SAEBench [1]. We will clarify them in the camera-ready version. The metrics are defined as follows:
>
> 	- The splitting metric counts how many latents are needed to represent a single feature. We measure this by training k-sparse probes and detecting when increasing k by one causes a jump in F1 score by more than threshold τ = 0.03. This indicates that additional latents contain significant information about the feature, suggesting the feature has been split across multiple latents.
>
> 	- The absorption rate measures the fraction of tokens where a latent corresponding to a first-letter feature fails to activate despite the token starting with that letter due to the feature being absorbed by another latent. First, we find false-negative tokens where the main SAE latents fail to fully activate on tokens starting with that letter (their projection onto the ground truth direction is less than the model's projection) while other latents (the absorbing latents) compensate on these tokens. Since we use the SAEBench implementation for our evaluation, we refer to their paper for more details [1].
>
> Thanks again for your valuable suggestions. We hope these clarifications and our planned revisions address your concerns. Given your positive assessment of our work's contribution and importance, we would be grateful if you would consider strengthening your support.
>
> [1] https://www.neuronpedia.org/sae-bench/info#feature-absorption
>
> [2] Chanin, David, et al. "A is for absorption: Studying feature splitting and absorption in sparse autoencoders." arXiv preprint arXiv:2409.14507 (2024).

---

### Decision · Program_Chairs · 2025-05-01

**Decision:**

Accept (poster)

**Comment:**

This paper proposes Matryoshka Sparse Autoencoders (Matryoshka SAEs), a new training method for sparse autoencoders designed to address interpretability issues such as feature splitting, feature absorption, and feature composition. The method is inspired by Matryoshka representation learning and introduces a nested structure, where the SAE is trained to reconstruct input features at multiple latent sizes. This encourages coarse features to be captured in smaller latent subsets and finer-grained features in larger ones. The approach is evaluated through experiments on synthetic data, a small transformer trained on TinyStories, and the Gemma-2-2B language model.

### Strengths

- Addresses an important problem in SAEs by targeting interpretability issues at scale rather than only focusing on sparsity vs. reconstruction trade-offs
- Novel and intuitive objective that enforces multi-scale feature learning through nested reconstruction constraints
- Demonstrates improved interpretability and concept isolation compared to strong baselines such as TopK SAE and JumpReLU SAE
- Evaluations are performed on both synthetic and realistic language modeling tasks, including experiments on a 2B-parameter model
- Reviewers highlighted the paper as well-written, with clear experimental design and extensive supplementary material
- The approach is simple yet powerful, opening possibilities for future integration with advanced activation functions or architectures
- Publicly available visualization tool (latent viewer) enhances transparency and usability of the method

### Weaknesses

- Reconstruction quality is worse than some baselines, raising concerns about performance trade-offs
- Evaluation tasks in certain sections (e.g., first-letter recognition) are narrow and may not reflect broader applicability
- The toy model experiments use idealized settings (e.g., orthogonal ground truth, known latent count) that may not generalize well
- Lack of evaluation of hierarchical (parent-child) relationships in real-world datasets beyond the toy model
- Some methodological descriptions (e.g., truncated Pareto distribution, sparsity control) are unclear or underexplained
- Unclear how well the method scales to larger language models beyond Gemma-2B, due to limited compute-based evaluation
- Some metrics (e.g., in Figure 6) are not clearly defined, and more qualitative examples could improve clarity

Most concerns have been addressed by the authors during the rebuttal period. All reviewers agree that this paper tackles an important problem and would be a great addition to the ICML program. The AC agrees.